# Vitamin D Supplementation in the Czech Republic: Socioeconomic Determinants and Public Awareness Gaps

**DOI:** 10.3390/nu17162623

**Published:** 2025-08-13

**Authors:** Drahomira Holmannova, Jan Hodac, Lenka Borska, Eva Cermakova, Lenka Hodacova

**Affiliations:** 1Department of Preventive Medicine, Faculty of Medicine, Charles University, Simkova 870, 500 03 Hradec Kralove, Czech Republic; hodacja@lfhk.cuni.cz (J.H.); borka@lfhk.cuni.cz (L.B.); hodacoval@lfhk.cuni.cz (L.H.); 2Department of Medical Biophysics, Faculty of Medicine, Charles University, Simkova 870, 500 03 Hradec Kralove, Czech Republic; cermakovae@lfhk.cuni.cz

**Keywords:** vitamin D, Czech Republic, vitamin D supplementation, vitamin D knowledge

## Abstract

**Background:** Vitamin D deficiency is a worldwide health problem associated with various health complications. This study aimed to determine the prevalence of vitamin D supplementation in the Czech Republic, understand reasons for supplementation, and assess participants’ knowledge of vitamin D’s physiological effects. **Methods:** The study included 1812 participants representing the Czech population aged 15+, stratified by gender, age, and regional distribution. Data analysis was performed using SASD 1.5.8, using chi^2^ independence tests and regression analysis. **Results:** The results revealed that only 13.5% of the participants maintained regular year-round vitamin D supplementation, while 51.5% never supplemented. A slight increase in supplementation was observed during the pandemic (2020–2021). Supplementation patterns were influenced by socioeconomic factors including age, gender, marital status, income, employment, and education (*p* > 0.001). Regarding vitamin D knowledge, 67.5% and 65.6% of participants recognized its role in immune system modulation and bone health, respectively. There were minor misconceptions, with 1.4% believing that it aggravates allergies and 1.8% linking it to cancer. Knowledge levels varied with education and residence size. **Conclusions:** Despite the high prevalence of vitamin D deficiency in the Czech population, regular supplementation remains low, indicating the need for enhanced prevention programs and awareness campaigns.

## 1. Introduction

In recent years, the importance of vitamin D in maintaining optimal health has gained considerable attention. Synthesized primarily through skin exposure to sunlight, vitamin D plays a key role in calcium balance and bone metabolism. However, its physiological influence extends beyond skeletal health, contributing to immune function, cardiovascular protection, and neurocognitive performance [1,2]. Although sunlight is an important natural source of vitamin D, deficiency remains alarmingly prevalent throughout the world—even in regions with sufficient sunshine [3,4]. This paradox is explained by multiple behavioral, environmental, and physiological factors. People often avoid direct sun exposure due to concerns about skin cancer or photoaging, use sunscreens that block UVB rays, or cover most of their skin for cultural or health reasons. In addition, spending most time indoors—whether due to work, lifestyle, or climate—substantially reduces UVB exposure. Older adults and individuals with darker skin pigmentation also have a reduced capacity to synthesize vitamin D in the skin [5]. Beyond sunlight, vitamin D can also be obtained from dietary sources, including fatty fish (such as salmon or mackerel), liver, egg yolks, and fortified foods like milk, cereals, and plant-based alternatives. Nevertheless, typical dietary intake is often insufficient to meet the body’s needs, particularly in populations with limited sun exposure or specific dietary habits. Therefore, vitamin D supplementation may serve as a useful adjunct in preventing hypovitaminosis D. However, it is essential to stress that supplements should complement—not replace—a balanced, nutrient-rich diet [6].

Deficiency in vitamin D has been linked to numerous chronic conditions, including diabetes, cardiovascular and neurodegenerative diseases, certain cancers, autoimmune disorders, and higher all-cause mortality [7,8,9]. A global analysis by Cui et al. found that the prevalence of vitamin D levels below 30, 50, and 75 nmol/L was 15.7%, 47.9%, and 76.6%, respectively. In our previous retrospective study of more than 100,000 Czech individuals, we confirmed widespread vitamin D inadequacy in the Czech population regardless of age; only children under one year of age had a low prevalence of vitamin D deficiency. The prevalence of deficiency (vitamin D level < 50 nmol/L) in adults divided by age ranged from 33.2% to 51.8%. Interestingly, the prevalence of deficiency was also high in younger adults aged 16 to 30, at 43.5% [10].

Current guidelines consider serum vitamin D concentrations below 75 nmol/L as insufficient and levels below 25 nmol/L as deficient. A daily intake of 2000 IU is generally considered safe and effective for preventing hypovitaminosis D [11]. It is important to emphasize that excessive supplementation of vitamin D may have adverse effects. Intakes exceeding 100 micrograms (4000 IU) per day may lead to vitamin D toxicity, which can result in hypercalcemia, vascular and tissue calcification, hypercalcuria, kidney damage, and other complications. Therefore, supplementation should always follow established recommendations and individual needs, ideally under medical supervision [12]. However, official recommendations state lower recommended daily values for vitamin D. According to the European Food Safety Authority (EFSA), the adequate intake (AI) for vitamin D is 15 µg/day (600 IU/day) for healthy individuals over one year of age and 10 µg/day (400 IU/day) for infants aged 7–11 months. The tolerable upper intake level (UL) is set at 100 µg/day (4000 IU/day) for adults and adolescents and 50 µg/day (2000 IU/day) for children aged 1–10 years [13].

The objective of this study was to identify patterns in vitamin D supplement intake and knowledge about vitamin D functions in the Czech population. The rationale for this investigation is based on the lack of comprehensive data regarding vitamin D supplementation behavior and related public knowledge within the general Czech population, particularly considering socioeconomic and demographic factors. To fill this gap, we conducted a representative cross-sectional population-based survey.

Unlike previous research, our study uniquely integrates the analysis of supplementation prevalence, motivational factors—including potential impacts of the COVID-19 pandemic—and public awareness of vitamin D’s physiological effects. The temporal focus on the period 2020–2021 is particularly relevant, given increased public attention to vitamin D’s role in COVID-19 outcomes [14].

Considering the high prevalence of vitamin D inadequacy in the Czech Republic [10], identifying at-risk population groups is essential for developing targeted public health strategies. Therefore, this study aims to provide actionable insights for evidence-based public health interventions and educational initiatives to improve vitamin D status, particularly among high-risk groups.

## 2. Materials and Methods

### 2.1. Study Design and Participants

The research plan and research design were developed during September–October 2023.

The field survey was conducted using the technique of a structured, standardized interview between the interviewer and the respondent (face-to-face). The final version of the questionnaire was determined based on the results of the pre-survey (Appendix A). Data collection was conducted by 207 professional interviewers from the INRES Agency across the Czech Republic. Optical and logical control, coding and data entry into the computer, and tabulation and interpretation of results were carried out by INRES Agency staff (the study was conducted in accordance with the Declaration of Helsinki, and the protocol was approved by the Ethics Committee of the Faculty Hospital in Hradec Kralove, Czech Republic (project identification code PROGRES Q40-09 and Q40-10, reference number 201705 I83P, date 2 May 2017)).

During the survey, a total of 1977 randomly selected citizens were approached by the interviewers with a request for an interview on the issue of health care and healthy living of the population. A total of 165 respondents, that is, 8.3% of all respondents, refused to be interviewed. On the other hand, 1812 respondents, that is, 91.7% of those questioned, agreed to be interviewed.

The data analyzed in this report were obtained from a sample of 1812 individuals selected through quota sampling. The sample is a representative sample of the population of the Czech Republic aged 15 and older. Representativeness was derived from the basic population of the Czech Republic aged 15 years and older. The size of the sample corresponds to a confidence level of 95%, and the margin of error is 3% (according to Raosoft SurveyWin verze 4.2). The degree of concordance (tightness) with the population is determined by the selection method and the applied sampling step. A two-stage selection was used: In the first stage, electoral districts were randomly selected from the basic set of electoral districts in the Czech Republic (quota region), and within them, individuals were randomly selected according to quotas (gender, age) to meet these quotas so that the selection corresponded to the structure of the basic set. In this way, a reserve was created in the sample, from which the parameters of the individual groups selected according to the quotas were balanced (Figure 1).

### 2.2. The Questionnaire

The questionnaire is available in the Appendix A. The survey contains six questions related to the use of vitamin D supplements, including the regularity of supplementation, and questions testing knowledge of the function of vitamin D in the human body.

### 2.3. Statistics

Statistical data processing was performed with SASD 1.5.8 (Statistical Analysis of Social Data). In the first stage of classification, frequency tables were constructed for individual indicators, and absolute and relative frequencies and mean values (mode, median, average, variance, standard deviation, range, estimate of variance and standard deviation, and interval estimate of mean and variance for 0.05) were calculated. In the second stage of classification, contingency tables with absolute and relative frequencies (column, row, total, and expected) and a sign scheme were constructed. As part of the correlation analysis, the Chi-square goodness-of-fit test—X^2^ (Pearson Chi-Square) was applied according to the nature of the distribution of the variables and the number of observations, followed by the Test of Independence. Furthermore, calculations of the Pearson contingency coefficient, the Normalized Pearson contingency coefficient, the Ćuprov coefficient, the Cramer coefficient, the Walis coefficient, the Spearman coefficient, and the Correlation coefficient were performed according to the nature of the variables and their distribution. The strength of the relationship was measured at three levels of significance (0.05, 0.01, and 0.001).

In the context of describing the analyzed statistically significant associations, the values of the Chi-square goodness-of-fit test (Pearson Chi-Square—X^2^) and the Test of Independence are standardly reported. To determine the direction of the statistically significant relationship between two variables, the possible deviation was calculated for each cell of the contingency table. Univariate logistic regression was also performed.

## 3. Results

### 3.1. Demographic Data

A total of 1812 people participated in our investigation. The distribution of our participants by age and sex is shown in Table 1. The distributions of participants corresponded to the distribution in the population of the Czech Republic according to the Age Structure of the Population—2022 document (status as of 31 December 2022, Prague, Czech Statistical Office 2023) [15]. Compared to the age distribution of the basic sample, the deviation does not exceed 0.1%. It can be stated that the research results are representative of the individual age groups of the population of the Czech Republic aged 15 and older (Table 1). The participants were also divided according to socioeconomic parameters (Table 2).

### 3.2. Intake of Vitamin D Supplements and Association Between Supplementation and Sociodemographic Data

More than 50% of the participants do not take vitamin D supplements, and only 13.5% of the participants take vitamin D regularly (every day), regardless of the season. The rest took it irregularly throughout the year, regardless of the season; irregularly during winter and spring when there is less sunshine; or regularly during winter and spring (Figure 2).

### 3.3. Dependence of Vitamin D Supplementation on Socioeconomic Factors

Statistical results, including univariate multiple logistic regression, used the use of vitamin D supplements as a dependent variable (answer 1—regular supplementation, and answers 2, 3, 4—irregular supplementation). The reference category for the dependent variable was answer 5 (people who did not take vitamin D supplements). The independent variables include gender, age, occupation, marital status, income, education, and place of residence. All parameters, except for the size of the place of residence, influenced vitamin D supplementation.

Due to regression analysis, gender had a statistically significant effect on supplement use (*p* < 0.001). Women had a 2.47 times higher chance of taking vitamin D supplements regularly and 1.82 times higher irregularly compared to men (OR = 2.47, 95% CI: 1.84–3.31 and OR = 1.82, 95% CI: 1.48–2.23). Age was a weaker predictor of supplement use (R^2^ = 0.0699), and only individuals aged 65 and older had a significantly higher chance of taking supplements regularly (OR = 2.49, 95% CI: 1.22–5.08) compared to younger individuals (15–19 years). Other age groups did not show significant differences. The effect of residence was minimal (R^2^ = 0.0275). Occupation was also a strong predictor (R^2^ = 0.14, *p* < 0.001). Individuals in occupation category 2 (mental worker: scientist, doctor, teacher, priest, actor, etc.) were 2.2 times more likely to take supplements irregularly and 1.47 times more likely to take supplements regularly (OR = 2.2, 95% CI: 1.17–4.12 and 1.47; 95% CI: 0.7–3.09), while those in occupations such as laborer, worker, farmer, armed forces employee, engineer, people without a regular salary (unemployed, students, home stay parents) were significantly less likely to supplement vitamin D. Self-employed individuals and entrepreneurs supplemented vitamin D the least, both in terms of regular and irregular supplementation (OR regular supplementation: 0.29; OR irregular supplementation: 0.67). Income has a positive effect on supplement use. Higher income groups (>60,000 CZK) had the highest chance of taking supplements (OR 1.87); however, in persons with an income of 20,001–30,000 CZK, the chance of taking vitamin D was also quite high (OR = 1.38, 95% CI: 0.76–2.53).

Marital status had a moderate effect (R^2^ = 0.0652, *p* < 0.001) on supplementation. Widowed individuals had a 2.68 times higher chance of taking supplements regularly (OR = 2.68, 95% CI: 1.61–4.46), and married individuals had a 1.97 times higher chance (OR = 1.97, 95% CI: 1.34–2.88). Education was a significant predictor (R^2^ = 0.0921, *p* < 0.001). Individuals in the highest education category (bachelor’s or higher degree) were 2.18 times more likely to take supplements regularly (OR = 2.18, 95% CI 1.13–4.2), while lower education levels did not show a statistically significant effect (Table 3; Appendix A).

### 3.4. Trends in Vitamin D Supplementation, Pre-Pandemic and Post-Pandemic, and Association with Socioeconomic Factors

Of the 880 participants who supplemented with vitamin D, almost 57% were already taking vitamin D before the COVID-19 pandemic (before 2020). During the period 2020 to 2021, 26.1% of 880 people started vitamin D supplementation. A total of 3.5% of 880 participants supplemented with vitamin D only during the pandemic (2020–2021) (Figure 3).

### 3.5. Association Between Period (Before, During, and After the COVID-19 Pandemic) of Vitamin D Use and Socioeconomic Characteristics

We also found two associations between the initiation of vitamin D supplementation in the context of the COVID-19 pandemic. The timing/initiation of vitamin D intake depended on age and marital status (*p* ˂ 0.001). Individuals aged 15 to 19 years were more likely to report that they did not start vitamin D supplementation until after 2022. Individuals aged 20 to 24 years were more likely to initiate vitamin D supplementation during the COVID-19 pandemic (2020–2021). Singles more often stated that they began supplementing vitamin D only after 2022, while married people were more likely to supplement vitamin D before the pandemic (2019) (Table 4).

### 3.6. Reasons for Vitamin D Supplementation

More than 42% of the participants reported that they had started vitamin D supplementation at the time of their decision. More than 17% had vitamin D recommended by a physician (Figure 4).

### 3.7. Association Between Reasons for Vitamin D Supplementation and Sociodemographic Data

We found an association between the reasons for vitamin D supplementation and all observed sociodemographic parameters (the null hypothesis was that socioeconomic factors do not influence the reason why respondents supplement vitamin D).

Men were more likely than women to report supplementing with vitamin D as a result of an education program on the positive effects of vitamin D on health. Women were more likely to have received vitamin D from a physician for a diagnosed deficiency.

Single students, participants aged 15 to 19 years, and respondents living in municipalities with 500 to 1999 inhabitants were significantly more likely to report taking vitamin D as a recommendation of family, friends, and acquaintances. Those who lived in municipalities of 2000 to 4999 inhabitants were significantly more likely to take vitamin D at their own discretion.

Widowed participants, participants with the lowest monthly household income (up to CZK 30,000), economically inactive citizens except students (retired, stay-at-home, on parental leave, or unemployed), respondents in larger cities (20,000 to 99,000 inhabitants), and participants 65 and older are significantly more likely to report being prescribed vitamin D by a physician based on an identified vitamin D deficiency. Divorced participants and employees are significantly more likely to report taking vitamin D based on advertising information.

College-educated respondents were significantly more likely to report taking vitamin D as a result of an educational program that informed them about the importance of vitamin D. Those with a high school education or without a high school diploma were significantly more likely to report that vitamin D was prescribed by a physician based on an identified vitamin D deficiency or recommended by a pharmacist (Table 5).

### 3.8. Participants’ Knowledge of the Importance of Vitamin D for Human Health

Respondents chose from nine options for the effects of vitamin D. Four were incorrect, and five were correct. A total of 67.5% and 65.6% of the respondents correctly stated that vitamin D positively modulates immune function and bone health, respectively. Only 21.4%, 20.7%, and 11.6% of the respondents knew that vitamin D has a positive effect on the cardiovascular system and the nervous system and decreases the risk of metabolic diseases. A total of 1.9%, 1.8%, 1.4%, and 1.3% of respondents incorrectly answered that vitamin D worsens skin condition, increases the risk of cancer, worsens lung function, and increases the risk of hypersensitive reactions (Figure 5).

### 3.9. The Association Between Knowledge About the Effect of Vitamin D on the Human Body and Sociodemographic Characteristics

We found an association between opinion on the function of vitamin D in the human body and gender, and the respondent’s place of residence. Men were significantly more likely to incorrectly report that vitamin D increases the risk of most cancers; similarly, residents of smaller cities (5000 to 19,999 inhabitants). However, residents of large cities (100,000 or more inhabitants) are significantly more likely to correctly report that vitamin D supports proper immune system functioning (Table 6).

## 4. Discussion

Vitamin D deficiency is a worldwide recognized public health concern, prevalent even in regions with high levels of exposure to sunlight and affecting both children and adults [4,16,17]. In the context of the European population, data obtained from the Netherlands, Germany, Ukraine, and the Czech Republic demonstrate that the prevalence of vitamin D deficiency exceeds 50%, especially in the elderly [10,18,19,20].

Ensuring adequate vitamin D status requires a multifaceted approach involving lifestyle, diet, and—where necessary—supplementation. Natural skin synthesis through sun exposure remains the most efficient source of vitamin D. According to Engelsen, a single full-body UV exposure, resulting in a slight pinkness in the skin (one minimum erythemal dose, 1 MED), is equivalent to an oral intake of 250–625 μg (10,000–25,000 IU) of vitamin D3 [21]. According to the findings of the Swiss study by Religi, during the summer and spring months, when 22% of the skin is exposed, adults can synthesize 1000 IU of vitamin D in 10 to 15 min of sun exposure [22]. Vitamin D production in the skin depends on factors such as skin pigmentation, age, latitude, and season. In the Czech Republic, solar radiation is insufficient for effective vitamin D synthesis. There are significantly fewer hours of sunlight annually (approximately 1670 h) compared to tropical countries such as Australia (approximately 3600 h) (data from main meteorological institutes: http://www.bom.gov.au/ and https://www.chmi.cz, accessed on 25 June 2025), particularly during autumn, winter, and early spring. This geographic limitation restricts the potential for cutaneous synthesis of vitamin D. Consequently, adequate intake through diet or supplementation becomes critical [21].

Dietary intake also contributes to total vitamin D levels. Among natural dietary sources, oily fish such as salmon, trout, and swordfish offer the highest vitamin D content (291–661.5 IU per 100 g), according to the United States Department of Agriculture [23].

Given vitamin D’s fat-soluble nature, it is stored in adipose tissue, and its serum concentration reflects both recent intake and long-term reserves. Therefore, maintaining a stable serum level—not merely daily intake—is essential for long-term sufficiency.

A nutrition survey within the WHO European region (2004 to 2015) revealed regional differences in vitamin D intake. Residents of the northern regions of Europe obtain approximately 7.28 μg/day of vitamin D from food, while residents of the western, central, and eastern regions of Europe obtain 3.65, 3.2, and 1.3 μg/day, respectively. There is a lack of data on vitamin D supplementation in the European Union [24,25]. Bischofova et al. focused on vitamin D intake in the Czech Republic. The range of daily vitamin D intake was 2.5–5.1 μg/day. The highest intake was in men aged 18 to 64 years, and the lowest intake was in children aged 4 to 6 years and girls aged 11 to 17. More than 95% of the Czech population has a vitamin D dietary intake below the recommended reference values of the diet [26]. A similar situation was described by Stachoń et al., who evaluated dietary vitamin D in Polish adolescents [27]. When there is insufficient sunlight and insufficient dietary intake of vitamin D, it is essential to supplement vitamin D levels with dietary supplements.

Given these findings, we conducted a population-based survey to evaluate the prevalence and patterns of vitamin D supplementation in the Czech Republic, as well as to assess public knowledge regarding its physiological roles. Understanding these trends is essential to designing effective prevention strategies targeting both the general population and specific high-risk groups. The question of how to ensure sufficient levels of vitamin D in the population is also being addressed, for example, in Germany or Denmark [28,29].

Despite the known benefits and widespread deficiency of vitamin D, our results reveal a strikingly low prevalence of regular year-round supplementation. Only 13.5% of respondents reported daily intake. Conversely, more than half (51.5%) of participants reported never taking vitamin D supplements. Other studies also show that vitamin D supplementation is not widespread among different populations in various countries. For example, Sha et al. examined UK Biobank data from 445,610 individuals and found that the rate of vitamin D supplementation in the UK is significantly lower than in the Czech Republic, at 4.3% [30].

However, the results of a smaller UK study by Tanna et al. based on a questionnaire survey of 557 people showed that nearly 50% of participants took vitamin D supplements, which deviates from the results of other studies [31].

A study conducted in Greece by Paparodis et al. revealed that more than 80% of the participants (5552 out of 6912) had never administered vitamin D supplements. A noteworthy finding was that only 67.9% of subjects who supplemented with vitamin D had sufficient levels of the nutrient [32].

Interestingly, a very high prevalence of vitamin D supplementation was documented in the Scottish study by Zaramba et al. In their study, 64.3% of 403 participants took vitamin D supplements, of which 37.5% took them daily during winter months [33].

These differences may reflect not only public health strategies, but also broader cultural norms and individual risk perception as well as several sociodemographic factors, which were confirmed by our results.

Regular and irregular supplement use was more prevalent among women, older individuals, married or widowed persons, those with higher educational attainment, and those with mental or administrative occupations. Interestingly, individuals from lower-income households were more likely to report supplement use—possibly reflecting physician-prescribed supplementation for diagnosed deficiencies in this group, particularly among older respondents. On the other hand, higher household income was also associated with a willingness to supplement vitamin D.

The present study is consistent with the findings of other research, which also indicates that women are more inclined to use supplements.

Mishra et al. found that women take dietary supplements of any kind more often than men (63.8% vs. 50.8%). They also described the association between age and vitamin D supplementation. In the group of people aged 20 to 39 years, 6.7% were supplementing with vitamin D, and in the group of people aged 60 years and older, 36.9% were supplementing. This result also corresponds with ours: older age is associated with a greater willingness to take vitamin D [34]. The association between female sex, age, and vitamin D supplementation was also confirmed by McCormack. The highest rate of supplementation was among women aged 65 to 69 (48%), while among men, it was highest in the 70–74 age group (23.5%) [35].

When it comes to education, many studies, including ours, have confirmed that education influences the willingness to supplement vitamin D.

Higher education is likely to be associated with greater knowledge of the impact of vitamin D on health and the prevalence of deficiency. Various studies that monitored the use of different dietary supplements have confirmed the link between education and supplement intake [36,37]. Studies also show that educational intervention increases the willingness to take dietary supplements, even when children are educated, bring this information into the family, and educate their parents [38,39]. Zaremba et al. showed that individuals with a college education demonstrate a higher likelihood of attaining average or above-average knowledge regarding vitamin D compared to those with less formal education, a finding that aligns with the outcomes of our study.

Our previous study used data from people before the COVID-19 pandemic, i.e., up to 2019. During 2020–2022, the importance of vitamin D was emphasized in the media and by experts, in the Czech Republic also by the National Health Information Portal (NZIP), which is under the Ministry of Health, in the article: COVID-19: dietary supplements during the pandemic which was written by the National Institute of Public Health [40]. It has been reported to reduce the severity of the disease course [41,42]. This led us to believe that this period may have motivated the population to take vitamin D. Therefore, we asked people if they took vitamin D before 2020, started supplementation during the period, or after 2022. Between 2020 and 2021, 26.1% of the 880 vitamin D supplementers in our study started taking vitamin D. A total of 3.5% of the 880 participants supplemented only in the pandemic years (2020–2021). Therefore, pandemics and awareness of the importance of vitamin D were able to increase the prevalence of vitamin D supplementation.

Participants cited a variety of reasons for initiating supplementation. Most often, individuals began supplementing on their own initiative or upon the recommendation of family or friends. A smaller proportion reported physician-initiated supplementation based on diagnosed deficiency. Educational campaigns, pharmacist recommendations, and advertising also contributed, though to a lesser extent. The reason for supplementation varied significantly across sociodemographic groups. For example, those with higher education were more likely to report participation in educational programs, while those with lower income or less education often cited physician recommendations.

We also surveyed the knowledge about the importance of vitamin D, specifically whether the participants knew what processes in the body affect vitamin D. It is this knowledge or lack thereof that may influence willingness to supplement vitamin D, so this is one factor that may influence it. While most respondents correctly identified its role in bone and immune health, knowledge of its effects on cardiovascular and neurological systems, as well as metabolic diseases, was limited. Alarmingly, a small proportion of the participants believed that vitamin D could have harmful effects, such as increasing the risk of cancer or allergies and worsening respiratory and skin conditions. These misconceptions were more frequent among men and residents of smaller towns.

A review of the extant literature revealed numerous studies that also focused on the issue of vitamin D knowledge. Notably, the results of several studies from other countries show that knowledge about the effects of vitamin D, including its role in bone health, is not high and is often lower than in the Czech population.

The Pakistan study by Tariq et al., with 900 participants, revealed that only 9% of the participants were able to identify appropriate dietary sources of vitamin D, and 33% were aware of the benefits of vitamin D for bone health. In our study, more than 65% of the participants knew the effect of vitamin D on bone tissue [43].

Alkalash et al. conducted a study in Saudi Arabia with 466 participants (67.8% had university education). Only 17.4% of them knew that the main source of vitamin D is sunlight exposure. It can be assumed that knowledge of the effects of vitamin D would also be poor [44]. In contrast, the results of the Polish study by Zadka et al [45]. exhibited a high degree of similarity to those of our own.

The results showed that 66% of 783 participants declared that vitamin D is necessary for bone and teeth health, but only 19% and 8% knew that vitamin D positively influences the activity of the immune system and nervous system. In our study, more than 67% of participants were aware that vitamin D has a positive effect on the functioning of the immune system [45]. Alibrahim et al. investigated awareness of the benefits and sources of vitamin D and the consequences of vitamin D toxicity. The positive effect of vitamin D on bone tissue was well known; however, the other information (the sun does not cause vitamin D poisoning, and being a vegetarian is more likely to be associated with vitamin D deficiency) was known by just over 54% [46]. Grużewska et al. conducted a survey on vitamin D use and knowledge in Poland with 203 participants. They found that 47.3% of the participants were taking vitamin D supplements, but 76% did not know the dose they were taking. A total of 70.3% of the respondents reported that vitamin D was related to bone health, which is a higher number compared to our study [47]. Participants in the study by Zaremba et al. also had a higher awareness of the effect of vitamin D on bone health. Over 80% of participants were aware of this effect, and more than 70% knew about the effect of vitamin D on the immune system. This figure is also higher than in the Czech population.

The study from Ghana by Kwabena revealed that the levels of awareness, knowledge, attitude, and practice in relation to vitamin D were 61%, 56.9%, 63.7%, and 73.2%, respectively. The analysis showed that young adults (18–24 years) and those with a higher education were significantly more likely to have better knowledge, while lower education and rural origin were associated with poorer knowledge. These results are consistent with ours [48].

Thus, it appears that vitamin D supplementation and awareness of its effects are not fully sufficient across populations in different regions of the world. Therefore, there is an effort to design programs that will inform the population and health experts about these facts.

The Polish study by Zgliczynski et al. conducted a survey with 701 medical doctors aged 32.1 ± 5.3 and revealed that the mean vitamin D knowledge score was 6.8 ± 2.3 out of 13 points. Only 14% of the participants supplemented vitamin D all year long, 24% only in autumn and winter, and 25% monitored their vitamin D levels. Most participants (61%) did not recommend regular vitamin D supplementation to their patients [49]. Educating health professionals about the risks of vitamin D deficiency is important, as they have a significant opportunity to educate patients and influence their health. The association between vitamin D deficiency and chronic diseases and multimorbidity is confirmed in many studies [8,30,50,51,52].

Vitamin D sufficiency is important not only at the individual level but also at the population level. Increased morbidity, particularly the incidence of chronic diseases and multimorbidity associated with vitamin D deficiency, is negatively reflected in the burden on health and social care and has negative economic consequences with a large impact on the labor market [53,54,55]. It may even lead to a health care crisis [56]. Luengo-Fernandez focused on cardiovascular diseases in the European Union (EU) and showed that the cost of cardiovascular disease in the EU is estimated at EUR 282 billion per year. EUR 155 billion (55%) is attributable to health and long-term care, representing 11% of EU health spending. Productivity losses account for 17% (EUR 48 billion), while the cost of informal care is EUR 79 billion (28%). The cost of cardiovascular disease was EUR 630 per person [57]. Further research is needed to establish accurate and effective guidelines and recommendations for vitamin D use in the general population and at-risk populations, and to monitor the effectiveness of these guidelines over time.

Study limitations: to maintain the representativeness of the population in our study, which is determined by age, gender, and place of residence, it was not possible to perform, for example, a cluster analysis to find the intersection of multiple factors that play a role in the willingness to supplement with vitamin D.

Income was per household, not per person in the household. People do not like to disclose their income, at most in a range within the whole household, e.g., 40,000 to 50,000 CZK. Thus, income per household is not a continuous variable, and it is not possible, even knowing the number of persons in the household, to calculate income per person. Knowing income per person would be more accurate than income per household. We tried to simplify the model and came to the same conclusions as in the case of income per household, i.e., the use of vitamin D supplements is not income dependent.

In addition, the research tool—a standardized questionnaire with predefined response categories—was optimized for statistical analysis but lacked open-ended or qualitative components. As a result, it could not capture the full complexity of participants’ motivations, attitudes, or beliefs regarding vitamin D supplementation. Psychosocial variables such as health literacy, perception of personal risk, influence of the media or social networks, and reasons for refusing supplementation were beyond the scope of this survey. Future studies could benefit from combined methods (e.g., focus groups or interviews) that would explore these dimensions in more detail. The questionnaire also did not examine whether respondents had practical knowledge of dietary sources of vitamin D or understood the mechanism of endogenous synthesis through sunlight. It is therefore not possible to assess whether gaps in supplementation are due to lack of awareness, limited access, or behavioral preferences.

Finally, as with all self-reported data, responses may be influenced by memory bias or socially desirable bias, particularly when it comes to health behaviors such as dietary supplement use.

Despite these limitations, the study offers robust population-level insights and highlights important sociodemographic trends that can inform targeted public health strategies. Future research should aim to incorporate more nuanced tools that capture the behavioral and cognitive aspects of supplementation choices and health literacy in greater detail.

## 5. Conclusions

This study offers nationally representative data on vitamin D supplementation practices and public knowledge, which can support targeted public health strategies. Despite the high prevalence of vitamin D deficiency, only a small proportion of individuals reported regular, year-round supplementation. Our findings indicate that sociodemographic factors—including gender, age, education, marital status, occupation, and income—significantly influence supplementation behaviors and the reasons for initiating vitamin D use. Additionally, although most respondents had general knowledge about the importance of vitamin D for bone health and the immune system, awareness of its other physiological functions was limited, and some misconceptions persisted. In order to address the high prevalence of vitamin D deficiency, public health strategies should focus on raising awareness of the benefits of vitamin D, correcting misinformation, and promoting a healthy lifestyle with sufficient outdoor exposure to sunlight, consumption of foods containing vitamin D, and vitamin D supplementation (this is of particular importance in the Czech Republic due to lower levels of sunlight), particularly among at-risk groups, such as young men and individuals with lower levels of education. The education of health care professionals is equally important to ensure that consistent and evidence-based recommendations are communicated to the public. Future studies should consider more comprehensive assessment tools to explore additional psychosocial and behavioral determinants of vitamin D supplementation and knowledge.

## Figures and Tables

**Figure 1 nutrients-17-02623-f001:**
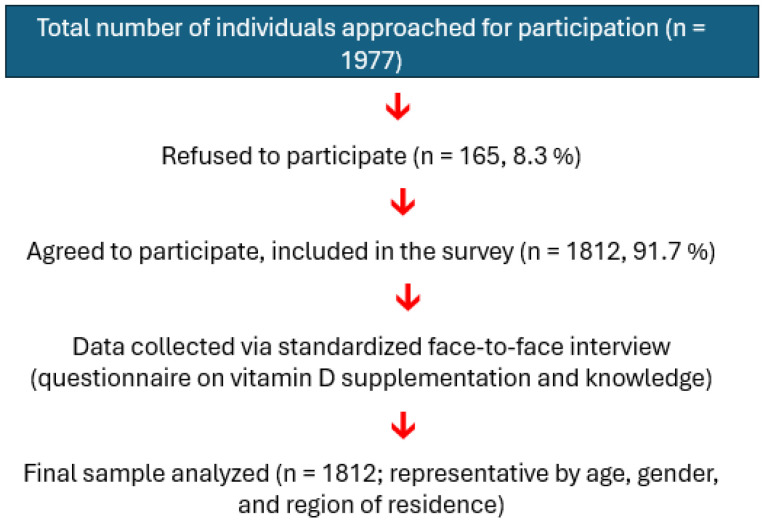
Flowchart describing participant recruitment and data collection in the survey on vitamin D supplementation and knowledge in the Czech population.

**Figure 2 nutrients-17-02623-f002:**
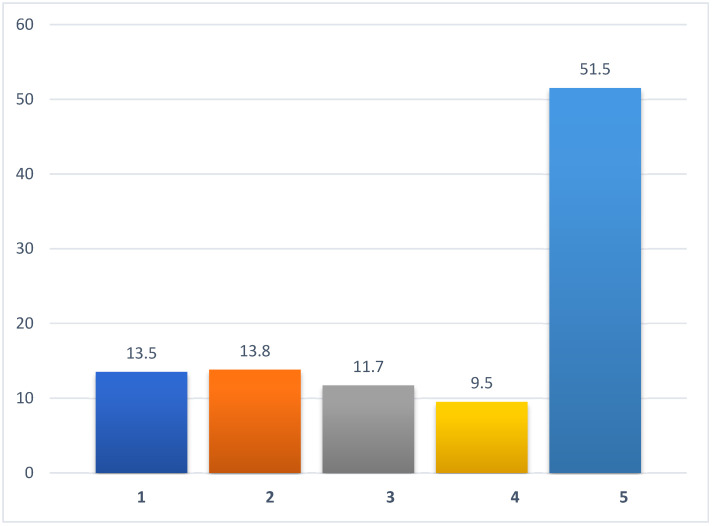
Regularity of vitamin D supplements (%, N = 1812). Legend: 1, regular, every day, use of vitamin D supplements; 2, seasonal use of vitamin D (winter, spring) every day; 3, irregular use of vitamin D throughout the year, regardless of the season; 4, irregular seasonal use of vitamin D (winter, spring); 5, no vitamin D supplementation. X-axis: vitamin D supplementation pattern. Y-axis: percentage of respondents (%).

**Figure 3 nutrients-17-02623-f003:**
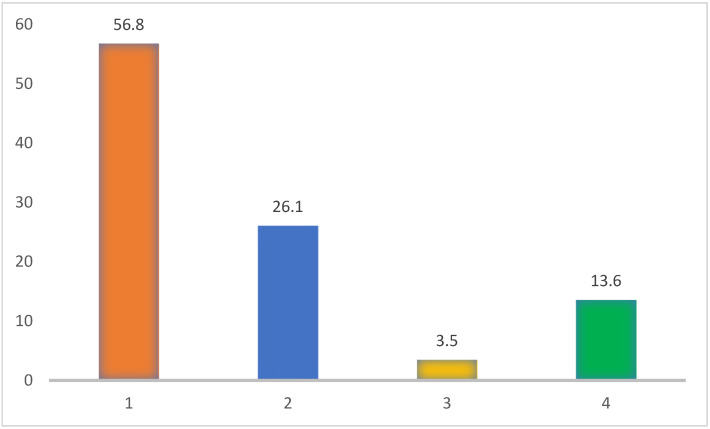
Vitamin D supplementation in relation to the COVID-19 pandemic (%; N = 880). Legend: 1—beginning of supplementation before 2020; 2—start of supplementation between 2020 and 2021 and still ongoing; 3—vitamin D supplementation only between 2020 and 2021; 4—start of supplementation after 2022. X-axis: timing of the supplementation. Y-axis: percentage of respondents (%).

**Figure 4 nutrients-17-02623-f004:**
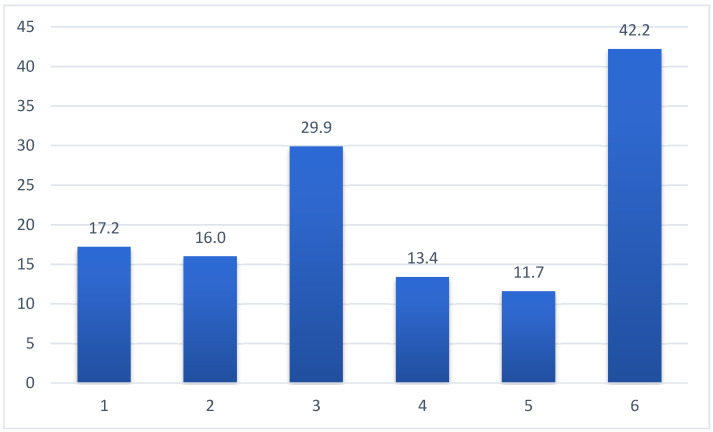
Reasons for vitamin D supplementation (%; n = 880). Legend: 1—vitamin D prescription by a physician due to a diagnosed vitamin D deficiency; 2—recommendation of vitamin D supplementation by a pharmacist; 3—recommendation of vitamin D supplementation by family members or/and friends; 4—recommendation of vitamin D supplementation based on information from advertising (TV, radio, Internet, outdoor advertising); 5—vitamin D supplementation based on educational and/or preventive program; 6—vitamin D supplementation on the decision of participants. X-axis: reasons for initiating vitamin D supplementation. Y-axis: percentage of respondents (%).

**Figure 5 nutrients-17-02623-f005:**
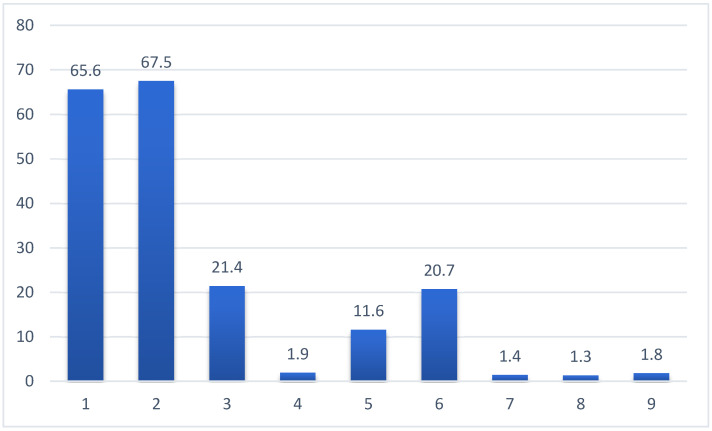
Views on the function of vitamin D in the human body (%; n = 1812). Legend: 1—helps maintain healthy bones (vitamin D deficiency is associated with a higher risk of osteoporosis); 2—supports the proper functioning of the immune system; 3—supports the proper functioning of the cardiovascular system; 4—worsens the skin condition (increases the incidence of acne and eczema); 5—reduces the risk of metabolic diseases (obesity, diabetes, and metabolic syndrome); 6—helps maintain the health of the nervous system, including brain function; 7—worsens the condition and functioning of the lungs; 8—increases the risk of allergies; 9—increases the risk of most cancers.

**Table 1 nutrients-17-02623-t001:** Participant data according to gender and age.

Years	Males			Females		
	N	%	Deviation	N	%	Deviation
**15–19**	58	3.2	0.1	56	3.1	0.1
**20–24**	52	2.9	0.1	51	2.8	0.1
**25–34**	133	7.3	0.1	126	7	0.1
**35–44**	157	8.7	−0.1	153	8.4	0
**45–54**	168	9.3	−0.1	165	9.1	0
**55–64**	126	7	−0.1	127	7	−0.1
**>65**	186	10.3	0.1	254	14	−0.1

Legend: N, number of participants.

**Table 2 nutrients-17-02623-t002:** Number of participants divided by socioeconomic parameters.

Occupation	Number
Manager, Director, Executive	87
Mental Worker	125
Armed Forces Employee	64
Engineer, Technician	94
Clerk, Administrative Worker	151
Service Worker	357
Farmer, Agricultural Worker, Forester	30
Laborer, Worker	109
Craftsman	56
Self-employed	47
Entrepreneur	33
Student	182
Homemaker, Parental leave	38
Retiree, Pensioner, Disabled Pensioner	426
Unemployed	13
Income	
0.0–20,000	129
20,001–30,000	291
30,001–40,000	354
40,001–50,000	308
50,001–60,000	269
60,001–70,000	198
70,001–80,000	263
Marital status	
Single	492
Married	859
Divorced	215
Widower, Widow	172
Partner, Common-law Partner	74
Education	
Primary/Elementary	121
Vocational Training or Secondary Education without a Diploma	512
High School Diploma, Higher Vocational Education	795
Bachelor’s Degree, University Degree	384
Total	1812

**Table 3 nutrients-17-02623-t003:** Association between vitamin D supplementation and sociodemographic data.

Vitamin D Supplementation	Value *X*^2^	*df*	*p* Value
**Gender**	55.92	4	<0.001
**Age**	38.81	24	˂0.05
**Marital status**	39.22	16	˂0.001
**Education**	51.90	12	˂0.001
**Size of place of residence**	17.96	20	0.590 n.s.
**Monthly household income**	24.13	8	˂0.01
**Occupation**	44.60	12	˂0.001

Legend: Pearson *X*^2^—chi square; *df*—degrees of freedom; n.s.—nonsignificant.

**Table 4 nutrients-17-02623-t004:** Association between period (before, during, and after the COVID-19 pandemic) of vitamin D use and socioeconomic characteristics.

	Value *X*^2^	*df*	*p* Value
**Gender**	3.89	3	0.27 n.s.
**Age**	43.96	18	˂0.001
**Marital status**	34.35	12	˂0.001
**Education**	15.37	9	0.08 n.s.
**Size of place of residence**	18.12	15	0.25 n.s.
**Monthly household income**	10.43	6	0.11 n.s.
**Occupation**	16.61	9	0.06 n.s.

Legend: Pearson *X*^2^—chi square; *p*—*p* value; *df*—degrees of freedom; n.s.—nonsignificant.

**Table 5 nutrients-17-02623-t005:** Association of reasons for taking dietary supplements or medications containing vitamin D with sociodemographic characteristics.

	Value *X*^2^	*df*	*p* Value
**Gender**	21.90	5	˂0.001
**Age**	84.28	30	˂0.001
**Marital status**	75.02	20	˂0.001
**Education**	41.42	15	˂0.001
**Size of place of residence**	46.20	25	˂0.01
**Monthly household income**	33.14	10	˂0.001
**Occupation**	80.65	15	˂0.001

Legend: Pearson *X*^2^—chi square; *df*—degrees of freedom, n.s.—nonsignificant.

**Table 6 nutrients-17-02623-t006:** The association between knowledge of the effect of vitamin D on the human body and socioeconomic factors.

	Value *X*^2^	*df*	*p* Value
**Gender**	20.28	8	˂0.01
**Age**	35.48	48	0.91 n.s.
**Marital status**	44.01	32	0.08 n.s.
**Education**	18.42	24	0.78 n.s.
**Size of place of residence**	64.26	40	˂0.01
**Monthly household income**	19.76	16	0.23 n.s.
**Occupation**	26.70	24	0.32 n.s.

Legend: Pearson *X*^2^—chi square; *df*—degrees of freedom; n.s.—nonsignificant.

## Data Availability

Data are available from the corresponding author upon reasonable request.

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
