# Peer review of "Vitamin D Supplementation in the Czech Republic: Socioeconomic Determinants and Public Awareness Gaps"

_nutrients, 2025, doi:10.3390/nu17162623_

Round 1
Reviewer 1 Report
Comments and Suggestions for Authors
The manuscript entitled “Exploring Supplementation Trends and Knowledge of Vitamin D Effects in the Czech Republic” presents interesting issues however some questions arise
- “Despite adequate sunlight in some regions, vitamin D deficiency remains alarmingly prevalent throughout the world” – It is necessary to clearly explain why sun exposure may be insufficient (e.g., avoiding the sun, use of sunscreen, covering the body for cultural or health reasons). It is also essential to add that vitamin D can be obtained from the diet, which was not mentioned at all. Instead, the need for supplementation was immediately emphasized. It should be stressed that dietary supplements are only a complement and not a substitute for a healthy diet. Therefore, the primary approach should be to obtain vitamin D through food sources.
- In our previous retrospective study of more than 100,000 Czech individuals, we confirmed widespread vitamin D inadequacy in the Czech population regardless of age, only children under one year of age have a low prevalence of vitamin D deficiency" - it is necessary to clarify what is meant by the prevalence of vitamin D levels and provide specific data.
- “Current guidelines consider serum vitamin D concentrations below 75 nmol/L as insufficient, and levels below 25 nmol/L as deficient. A daily intake of 2,000 IU is generally considered safe and effective for preventing 50 hypovitaminosis D [8].” - It is essential to reference official recommendations rather than individual publications. Only guidelines issued by scientific societies or international institutions should be cited, as there are numerous review articles that may not reflect official consensus. (eg. https://www.nhs.uk/conditions/vitamins-and-minerals/vitamin-d/; EFSA, Institute of Medicine. Dietary reference intakes for calcium and vitamin D. Washington, DC: The National Academies Press; 2011. Etc.). Additionally, it should be specified which age group the recommendations apply to, as guidelines vary depending on age.
- In the introduction, it is essential to include information about the potential harmful effects of excessive vitamin D supplementation (more than 100 micrograms [4,000 IU] per day may be harmful). (please see https://www.mayoclinic.org/healthy-lifestyle/nutrition-and-healthy-eating/expert-answers/vitamin-d-toxicity/faq-20058108; https://www.health.harvard.edu/staying-healthy/taking-too-much-vitamin-d-can-cloud-its-benefits-and-create-health-risks; https://pmc.ncbi.nlm.nih.gov/articles/PMC10195747/)
- “The sample is a representative sample of the population of the Czech Republic aged 15 and older.” – It should be clarified why minors were included in the study, and how consent for their participation was obtained from their legal guardians or parents
- Line 91 “The Questionnaire” – The name of the questionnaire, its validation, and other relevant details should be described here. It is not sufficient to simply include the questionnaire as a supplement. The research tool must be properly described in the main text.
- “In case of insufficient observations, Yates's correction was 113 applied.” – Could you please define what is meant by 'insufficient observations'? Specifically, what threshold or criteria were used to determine when Yates's continuity correction was necessary in your statistical analysis? Is it referring to 2x2 tables only, or also to small sample sizes?
- “1812 people participated in our investigation” You shouldn’t start a sentence with a number. (This sentence can be rephrased as follows: “A total of 1812 people participated in our investigation.”
- Figures 1 are unnecessary. Data should be mentioned only in the text,. The data presented as a figure do not provide any additional value. Like the other figures, it is also unnecessary and does not need to be presented to show the data.
- Please correct the captions under the tables because there are some typos and inaccuracies
- In discussion, the authors refer, among other things, to the prices of supplements, which is absolutely unnecessary. First, there should be references to the prevalence of vitamin D levels in other countries but in similar groups. The group studied in this article is quite broad, so it would be worthwhile to discuss this in subgroups. Second, there should be clear guidelines regarding the increase of vitamin D intake through diet, as well as practical information on how much time one should spend in the sun (and at what body exposure) to achieve the daily requirement. Vitamin D is stored in fat tissue, so daily intake or sun exposure alone are not the key; maintaining a stable blood level is what matters. All of this should be addressed; otherwise, the discussion looks like a commercial advertisement for buying and consuming supplements.
- The study limitations should be elaborated on in more detail, especially in the context of the research tool, which is simply a series of questions with predefined answers. The motivations or attitudes of consumers toward taking vitamin D supplements were not examined here, nor was whether consumers know which products contain vitamin D or how it can be obtained from sun exposure.
- “This study provides valuable information on attitudes, behaviors, and knowledge 49 related to vitamin D supplementation within the Czech population” - This sentence should be removed because the authors do not study attitudes toward vitamin D supplementation. Even in the case of behaviors and knowledge, the analysis is not comprehensive but rather quite limited. Conclusions should be formulated with greater caution.
Author Response
"Please see the attachment."

Reviewer 2 Report
Comments and Suggestions for Authors
This study explores the vitamin D deficiency situation among residents of the Czech Republic and calls for the general public to supplement their vitamin intake.
Please revise the title, as the current one does not reflect the content of the study.
Please add the rationale for conducting this research and its innovation in the introduction.
Please include the ethical approval number in the methods section.
Please specify the name of the ethical approval committee in the methods section.
Line 91-92: Please complete this sentence.
Please check the details, such as sentences missing periods at the end. Additionally, other punctuation format issues should also be addressed.
The discussion section contains excessive focus on vitamin D deficiency in different regions, but after reviewing the manuscript, I don’t believe this is a primary research variable for the authors. I think this part can be simplified, such as highlighting that vitamin D deficiency is more severe in the author’s region compared to other areas.
The manuscript lacks a flowchart; please add one.
Please compare your research findings with those of other studies you cited in the discussion, rather than simply presenting their findings.
Table: Please capitalize the first letters of the table headings.
Why are some words in the table all uppercase while others are all lowercase?
Please check the axes of the figures; I don’t understand what the x and y axes represent in some of the figures, as the authors did not provide this information.
Author Response
"Please see the attachment."

Round 2
Reviewer 2 Report
Comments and Suggestions for Authors
The manuscript is approved.